# Interaction of Nipah Virus F and G with the Cellular Protein Cortactin Discovered by a Proximity Interactome Assay

**DOI:** 10.3390/ijms25074112

**Published:** 2024-04-08

**Authors:** Chunmei Cui, Pengfei Hao, Chaozhi Jin, Wang Xu, Yuchen Liu, Letian Li, Shouwen Du, Limin Shang, Xin Jin, Ningyi Jin, Jian Wang, Chang Li

**Affiliations:** 1Changchun Veterinary Research Institute, Chinese Academy of Agricultural Sciences, Changchun 130122, China; cuichunmei97@163.com (C.C.); haopf19@mails.jlu.edu.cn (P.H.); xuwang@nwafu.edu.cn (W.X.); letian823@163.com (L.L.); du-guhong@163.com (S.D.); jinny@cae.cn (N.J.); 2Preventive Veterinary Medicine Laboratory of Agricultural College, Yanbian University, Yanji 133000, China; jinxin@ybu.edu.cn; 3State Key Laboratory of Proteomics, Beijing Proteome Research Center, National Center for Protein Sciences (Beijing), Beijing Institute of Lifeomics, Beijing 102206, China; jinchaozhi@ncpsb.org.cn (C.J.); liuyuchen@ncpsb.org.cn (Y.L.); data.cool@163.com (L.S.)

**Keywords:** Nipah virus, CTTN, viral entry, proximity labeling

## Abstract

Nipah virus (NiV) is a highly lethal zoonotic virus with a potential large-scale outbreak, which poses a great threat to world health and security. In order to explore more potential factors associated with NiV, a proximity labeling method was applied to investigate the F, G, and host protein interactions systematically. We screened 1996 and 1524 high-confidence host proteins that interacted with the NiV fusion (F) glycoprotein and attachment (G) glycoprotein in HEK293T cells by proximity labeling technology, and 863 of them interacted with both F and G. The results of GO and KEGG enrichment analysis showed that most of these host proteins were involved in cellular processes, molecular binding, endocytosis, tight junction, and other functions. Cytoscape software (v3.9.1) was used for visual analysis, and the results showed that Cortactin (CTTN), Serpine mRNA binding protein 1 (SERBP1), and stathmin 1 (STMN1) were the top 20 proteins and interacted with F and G, and were selected for further validation. We observed colocalization of F-CTTN, F-SERBP1, F-STMN1, G-CTTN, G-SERBP1, and G-STMN1 using confocal fluorescence microscopy, and the results showed that CTTN, SERBP1, and STMN1 overlapped with NiV F and NiV G in HEK293T cells. Further studies found that CTTN can significantly inhibit the infection of the Nipah pseudovirus (NiVpv) into host cells, while SERBP1 and STMN1 had no significant effect on pseudovirus infection. In addition, CTTN can also inhibit the infection of the Hendra pseudovirus (HeVpv) in 293T cells. In summary, this study revealed that the potential host proteins interacted with NiV F and G and demonstrated that CTTN could inhibit NiVpv and HeVpv infection, providing new evidence and targets for the study of drugs against these diseases.

## 1. Introduction

Nipah virus (NiV) belongs to the genus of the paramyxovirus family Henipavirus, which first broke out in Malaysia in 1998 and was subsequently detected in other countries such as Bangladesh, Singapore, and the Philippines [1]. NiV is a single-stranded, negative-stranded, non-segmental enveloped RNA virus. NiV consists of five structural proteins: nucleocapsid protein (N), matrix protein (M), glycoprotein (G), fusion protein (F), and long polymerase (L), and four non-structural proteins, P, C, W, and V [2,3]. It has a high mortality rate and is a biosafety level 4 zoonotic pathogen [1,4]. The main symptoms of NiV infection are fever, headache, cough, dyspnea, diarrhea, and, in some cases, encephalitis, and seizures [5].

The two surface glycoproteins of NiV, attached glycoprotein G and fusion glycoprotein F, are required for viral entry into host cells and are major determinants of NiV-infected species and host cell tropism. The G protein mediates attachment to cell receptors, while the F protein is responsible for the successful entry of the virus into the host cell by initiating pH-independent fusion of the virus and the cell membrane [6]. Unlike other paramyxoviruses, the NiV G protein does not exert the effects of hemagglutinin and neuraminidase [7]. Instead, the protein binds to the host cell receptors Ephrin B2 and B3 to enter the virus. As a type II membrane protein, G proteins exhibit characteristic tetramerization through their N-terminus α helical stalk domains, while their globular head domains located at the C-terminus bind to host cell receptors [8,9]. On the other hand, NiV F proteins come in the form of trimers and belong to class I viral fusion proteins. They have a globular head consisting of three domains and are attached to the host cell membrane by a C-terminal α helical stalk [10]. In addition, from the perspective of the immune response, NiV G and F proteins are particularly important as target antigens that can trigger the production of NiV-neutralizing antibodies, which presumably target the NiV receptor-binding protein head domain, thereby competing with the binding of NiV G to Ephrin B2/B3 and its receptors (Eph) and inhibiting viral entry.

Proximity labeling (PL) is used for proteomic analysis of macromolecular complexes, organelles, and protein interaction networks [11]. In PL, the target protein is fused with TurboID, a biotinylated enzyme that can catalyze endogenous proteins in a proximity-dependent manner, which can quickly and non-toxically label the protein interaction with the target protein or its proximal protein in just 10 min in the biotin-containing environment [12]. The high efficiency of TurboID has led to its application in the study of a variety of animal viruses, including SARS-CoV-2, hepatitis B virus (HBV), herpes virus, and rotavirus [7,13,14,15].

Pseudovirus refers to a virus that has an envelope of another exogenous virus but a retrovirus with a genome, and it is a class of chimeric virus particles with the genomic characteristics of a virus. It can accurately simulate the process of wild-type virus entering host cells and can also accurately determine the virus to the host. Pseudovirus-based entry assays have been widely used to elucidate viral invasion mechanisms [16].

Although the number of cases in the outbreak of NiV is relatively small compared to other viruses, it is highly likely to have a large-scale epidemic worldwide due to its human susceptibility, high mutation rate, and wide range of hosts. In addition, human invasion of the habitat of its host, the fruit bat, may also lead to more and more virus spillover, increasing the likelihood of NiV outbreaks and threatening the safety and health of humans and the development of the livestock industry. In addition, there is currently only one approved subunit vaccine for henipavirus, and most treatable drugs are only allowed during outbreaks, such as ribavirin, a purine nucleoside analog that inhibits viral RNA polymerase, which was used during outbreaks in 1998 and 2018 [17,18]. Therefore, in order to develop therapeutic drugs to defend against NiV, it is necessary to explore the infection mechanism of NiV, in particular, it is important to explore more host factors associated with NiV infection. The main purpose of this study was to sift out host proteins that have potential interactions with NiV F and G by proximity labeling, to lay a data foundation for the study of the mechanism of NiV infection. In this study, we used TurboID PL combined with mass spectrometry (MS) to study the protein–protein interactions (PPIs) of NiV F and G with the host proteins. A total of 1996 and 1524 host proteins were identified as interacting with NiV F and G proteins, respectively. In addition, through further validation of PPIs, we found that CTTN interacts with both NiV F and NiV G and inhibits the infection of NiVpv into host cells. The findings provide crucial information for better understanding the mechanism of NiV infection and the development of vaccines or antiviral drugs to defend against NiV.

## 2. Results

### 2.1. TurboID Proximity Labeling Combined with LC-MS/MS Reveals NiV F/G-Host Protein Interactomes

TurboID proximity labeling combined with LC-MS/MS was used to screen the host proteins that have potential for interaction with NiV F and G proteins in 293T cells (Figure 1A). After filtering, a total of 1996 proteins interacting with NiV F and 1524 host proteins interacting with NiV G were obtained, and among them, 863 overlapping proteins interacted with both F and G (Figure 1B). Importantly, the NiV host receptor protein, EFNB2/EFNB3, is included in NiV G-interacting host proteins (Figure 1C); this discovery is consistent with a previous report, confirming the credibility of our data [8]. The NiV F/G-interacting host proteins were analyzed by Gene Ontology (GO) enrichment (Figure 1D,F). Most of these host proteins were involved in cellular processes and molecular binding. And, the proteins were selected for Kyoto Encyclopedia of Genes and Genomes (KEGG) analysis (Figure 1E,G); most of these host proteins were involved in endocytosis and tight junction. The 863 overlapping host proteins were analyzed by GO (Figure 1H) and KEGG enrichment (Figure 1I). The results showed that a majority of proteins were involved in cellular processes, molecular binding, endocytosis, tight junction, and other functions. 

### 2.2. CTTN, SERBP1, and STMN1 Were Distributed on the Cell Membrane with NiV F and G

The PPIs network diagram (Figure 2A) and Venn diagram (Figure 2B) show that only CTTN, SERBP1, and STMN1 were in the overlapping part, among the top 20 high-confidence proteins that interacted with NiV F and G proteins. Therefore, we decided to perform the further validation of these three proteins. To confirm the data, we used confocal fluorescence microscopy to observe the subcellular distribution of NiV F protein (Figure 2C), NiV G protein (Figure 2D), CTTN, SERBP1, and STMN1 in HEK293T cells. NiV F and G plasmids were transfected into HEK293T cells, each plasmid was transfected with 2.5 μg, and the immunolocalization of NiV F and G proteins and CTTN, SERBP1, and STMN1 in cells was carried out after 24 h. The results showed that bright yellow parts could be seen in the cells, which indicated the F protein and G protein, respectively, had a clear colocalization relationship with CTTN, SERBP1, and STMN1 in cells, and the results of ImageJ software (ImageJ 1.53e, Java 1.8.0_172) analysis showed that in the red fluorescence of the white line mark shown is almost coincident with the green fluorescence, which further indicated that the distribution of NiV F and G proteins was similar to that of CTTN, SERBP1, and STMN1 in cells. CTTN, SERBP1, and STMN1 have the possibility of direct interaction with NiV F and G proteins.

### 2.3. Overexpression of CTTN Restricted Infection of the NiVpv in HEK-293T Cells

To explore the function of CTTN, SERBP1, and STMN1 in infection of NiV, we conducted pseudoviral infection experiments. The Western blot results showed that the CTTN (Figure 3A), SERBP1 (Figure 3B), and STMN1 (Figure 3C) were expressed successfully after transfection for 24 h, and they had no significant effect on cell viability compared with control by CCK8 analysis (Figure 3D–F), indicating the transfection system could be applied to the following experiment. Subsequently, 2.5 μg of CTTN, SERBP1, and STMN1 plasmids were transfected into HEK293T cells, and pseudovirus infection experiments were carried out 24 h later. The luciferase detection results showed that overexpression of CTTN inhibited the infection of NiVpv significantly (*p* < 0.0001) (Figure 3G), while overexpression of SERBP1 (Figure 3H) and STMN1 (Figure 3I) had no significant effect on NiVpv infection (*p* > 0.05). To confirm the function of CTTN on NiVpv infection, a dose-dependent experiment was conducted. HEK293T cells were transfected with 0, 0.2, 0.6, and 1.8 μg CTTN plasmids, respectively, and pseudovirus infection experiments were performed after 24 h. Western blot showed that the expression of CTTN is increased with increasing transfection dose (Figure 3J). The infection titers of pseudovirus were detected by luciferase, and the results showed that the infection of NiVpv was inhibited with the increase in the expression of CTTN (Figure 3K). The results showed that CTTN could inhibit NiVpv infecting cells and presents dose dependence.

### 2.4. Loss of Function of CTTN Promoted Infection of the NiVpv

To further demonstrate whether CTTN inhibits NiVpv infection of host cells, a siRNAs experiment was performed, and 100 nM siRNAs were transfected into HEK293T cells. Western blot results showed that the No.3 siRNA of CTTN had the most significant interference effect and could be used for subsequent experiments (Figure 4A,B). CCK8 results also showed that transfection of siRNA for 24 h had no significant effect on cell viability (Figure 4C). Then, NiVpv was seeded into siRNA-treated cells and detected by luciferase, and the results showed that knockdown of CTTN promoted the infection of NiVpv in HEK293T cells significantly (*p* < 0.0001) (Figure 4D). The result also indicates that CTTN could inhibit the infection of NiVpv, which is consistent with the overexpression results.

### 2.5. The Infection of HeVpv Is Inhibited by CTTN

Using the same method as NiVpv (2.3–4), we also examined the effect of CTTN on HeVpv-infected HEK293T cells, and the results showed that overexpression of CTTN could inhibit the HeVpv infection of HEK293T cells significantly (*p* < 0.0001) (Figure 5A). Knockdown of the expression of CTTN promoted the infection of HeVpv significantly (*p* < 0.0001) (Figure 5B), and with the increase in CTTN expression, the infection of HeVpv was increasingly inhibited (Figure 5C).

## 3. Discussion

NiV is one of the most pathogenic zoonotic viruses that have recently spilled over from bats, causing serious health concerns, economic problems, and posing a risk of pandemic transmission. There is neither a licensed treatment nor a vaccine, so more research is needed to explore the mechanism of NiV infection. Previous research has found that EFNB2/B3 has been identified as the receptor of NiV, and a cholesterol, Griffithsin (GRFT), has also been found to be associated with the fusion of NiV and cell membranes [19,20,21,22].

As a member of the family Paramyxoviridae, NiV glycoprotein G binds to the receptor and triggers a conformational change in fusion protein F, which fuses with the cell membrane and thus enters the cell, completing the first step of viral infection [6].

The subcellular localization of viral membrane proteins in living cells and the molecular identity of adjacent or interacting proteins contribute to the search for infection-related factors for viral pathogens [23]. In this study, we used the TurboID proximity labeling method to efficiently label the host proteins that interact with NiV F and G proteins in HEK293T cells. Further analysis of the database found that the receptors EFNB2 and EFNB3 of NiV were included in high-confidence proteins that interact with NiV G proteins which is consistent with previous reports, proving that the protein database we screened is credible; therefore, it provides a strong database for subsequent research on the NiV mechanism [8]. Subsequently, we performed GO and KEGG enrichment analyses on these high-confidence protein databases, and the results showed that most cells were involved in endocytosis, molecular binding, and tight junction, which was consistent with the functions of NiV F and G proteins, increasing the possibility of these proteins interacting with NiV F and G proteins.

From the protein database, we found that CTTN, SERBP1, and STMN1 all existed in the top 20 protein ranks that interacted with NiV F and G proteins. Therefore, we selected these three proteins for further interaction verification. The colocalization results showed that these three proteins were in close proximity to NiV F and G proteins in HEK293T cells, indicating that CTTN, SERBP1, and STMN1 had the possibility of interaction with NiV F and G proteins. Next, we conducted pseudovirus infection experiments, and the results showed that CTTN could significantly inhibit the infection of NiVpv, while SERBP1 and STMN1 had no significant effect on the infection of pseudovirus. Previous studies have shown that SERBP1 is mainly involved in processes such as mRNA stability and regulation of post-transcriptional genes [24]. Gene ablation of SERBP1 reduces cellular sensitivity to DENV and related flaviviruses by hindering translation and replication steps [25]. Studies have shown that STMN1, as a microtubule depolymerization protein, affects HIV-1 infection but is involved in HIV-1 transcription [26,27]. We suspect that the reason why SERBP1 and STMN1 do not significantly affect NiVpv infection is due to the limitation that the pseudovirus we used can only carry out one round of infection and cannot replicate. As for how these two proteins interact with NiV F and G proteins—this needs to be further explored. CTTN was initially identified as a substrate for Src kinases and plays a key role in the formation of cell membrane protrusions such as lamellipodia and invasion foot, as well as in maintaining the integrity of adheresion junctions and cell-to-cell junctions [28]. In several past studies, CTTN has been found to be involved in the infection process of a variety of viruses. CTTN has been shown to promote viral infection in several virology studies, including the Influenza virus, Hepatitis C Virus (HCV), and Rotavirus (RV). In addition, in RSV, it has been found that CTTN can inhibit its infection by protecting the epithelial barrier [29,30,31,32]. In conclusion, further research is required on how CTTN interacts with NiV F and G proteins and how it is involved in viral infection. Since there is no BSL-4 laboratory in this experiment, only pseudoviruses can be used for infection experiments, which can only indicate that CTTN interacts with F and G and has the possibility of mediating the infection of NiV.

In conclusion, this study constructs a robust protein database for global drug discovery teams to develop therapeutic drugs to defend against NiV and demonstrates that CTTN is a potentially relevant host factor for the infection of NiV.

## 4. Materials and Methods

### 4.1. Cells and Viruses

HEK293T cells were purchased from BeNa Culture Collection, China. Cells were cultured in a Dulbecco’s modified Eagle’s medium (DMEM; Gibco, New York, NY, USA) containing 10% fetal bovine serum (FBS; Gibco) and 1% penicillin-streptomycin solution (Hyclone Laboratories, Logan, UT, USA).

The NiVpv and HeVpv used in the study were kindly gifted from Dr. Youchun Wang (Division of HIV/AIDS and Sexually Transmitted Virus Vaccines, National Institutes for Food and Drug Control (NIFDC), Beijing, China).

### 4.2. Plasmid Construction, siRNA, and Transfection

NiV F and G genes (GenBank: NC_002728.1) were cloned into pcDNA3.1-Myc-TurboID, generating the expression plasmid pcDNA3.1-Myc-TurboID-F and pcDNA3.1-Myc-TurboID-G, respectively. Human CTTN (GenBank: NM_005231.3) and SERBP1 (GenBank: AF151813.1) coding sequences were amplified from HEK293T. The STMN1 gene (GenBank: AB451319.1) was synthesized by Sangon Biotech. Then, all three genes were cloned to pcDNA3.1-3 × Myc vector by EcoRⅠ/XbaⅠ or Hind Ⅲ/XbaⅠ. All the ligation products were transformed into *Escherichia coli* Trans-T1 competent cells and plated on ampicillin-resistant agar plates, and single clones were picked and shaken in a 37 °C incubator. Plasmids were extracted and identified.

The cells were grown to 80% confluence prior to transfection, and plasmids were transfected with Lipofectamine 3000 (Thermo Scientific, Waltham, MA, USA) for 24 h according to the manufacturer’s instructions.

siRNAs of CTTN (cat# SIGS0002537-1), SERBP1 (cat# SIGS0010637-1), and STMN1 (cat# SIGS0004135-1) were purchased from RiboBio Co., Ltd. (Guangzhou, China). The cells were grown to 80–90% confluence prior to transfection, transfected with siRNAs using Lipofectamine RNAiMAX (Thermo Scientific, USA) for 24 h, and were then detected or used for further experiment.

### 4.3. Cytotoxicity Assay

The day before transfection, HEK293T cells were plated in 96-well plates and cultured at 37 °C. Then, pcDNA3.1-3 × Myc-CTTN, pcDNA3.1-3 × Myc-SERBP1, pcDNA3.1-3 × Myc-STMN1, and the siRNAs of these three molecules were transfected into cells and incubated at 37 °C for 24 h under CO_2_ conditions. Subsequently, Dilute cell counting kit-8 (Dojindo, Japan) with DMEM at a 10:1 ratio was used, cell supernatant was discarded, and 100 μL of mixed solution per well were added, placed in a 37 °C incubator and incubated for 1 h. Then, the optical density (OD) value was determined at 450 nm using the TECAN sunrise 96-well microplate reader.

### 4.4. Pseudovirus Infection Assay

HEK293T cells were plated into 12-well plates, transfected with overexpression plasmids or siRNAs, and pseudoviruses (a kind gift from Prof. Weijing Huang) were diluted with DMEM and added to 96-well plates. Then, cells diluted with DMEM were added (1 × 10^5^/well), incubated at 37 °C for 48 h, and detected with luciferase detection reagent (Vazyme, Nanjing, China) by the TECAN sunrise 96-well microplate reader, according to the instructions.

### 4.5. Proximity Labeling with TurboID Assays

As previously described, the constructed pcDNA3.1-Myc-TurboID-F and pcDNA3.1-Myc-TurboID-G were transfected into HEK293T cells. Then, biotin (50 μM) was added, and three replicates were set up, and experiments and analysis were conducted as previously described [33].

### 4.6. Western Blotting

Cells were collected, washed with PBS, and lysed with Western IP lysate (Beyotime, Shanghai, China) for 30 min. The supernatant was taken for BCA protein concentration determination, and then 5 × SDS was added, and the protein samples were boiled in water for 10 min, equal amounts of 30 μg were separated by 12.5% SDS-PAGE and transferred onto the NC membrane. The NC membrane was blocked with a Rapid blocking solution (Beyotime, China) at room temperature for 30 min, then incubated overnight at 4 °C with primary antibodies (1:1000). After further incubation with goat anti-rabbit IgG-HRP and goat anti-mouse IgG-HRP secondary antibodies (1:5000, purchased from Beyotime), the membranes were detected by the Amersham Imaging 600 system, with Pierce ECL Western blotting substrate (32106; Thermo Fisher Scientific).

### 4.7. Immunolocalization and Confocal Assay

Cells were fixed with 4% paraformaldehyde solution (P1110; Solarbio, Beijing, China) for 1 h. The cells were washed thrice and blocked with BSA for 1 h. After washing thrice, specific primary antibodies (1:500) were added and incubated overnight at °C. Cells were then washed and incubated with the goat anti-rabbit IgG-CY3 and goat anti-mouse IgG-FITC fluorescent secondary antibodies (Beyotime) for 1 h. Images were captured by a fluorescence or a confocal microscope.

### 4.8. Statistical Analysis

Statistical significance between groups was determined using GraphPad Prism 9.0 software (GraphPad Software Inc., San Diego, CA, USA). Data were presented as mean ± standard error of the mean (SEM) in all experiments and analyzed using a *t*-test or analysis of variance. *p* < 0.05 was considered statistically significant.

## Figures and Tables

**Figure 1 ijms-25-04112-f001:**
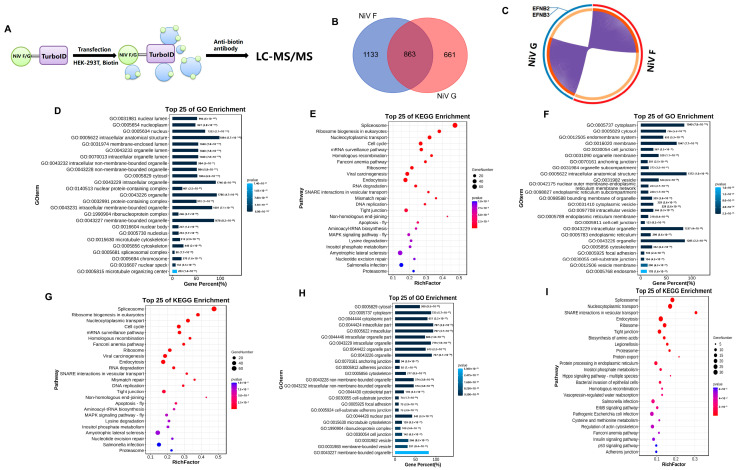
Bioinformatics analysis. (**A**) Flow chart of turboID proximity. (**B**) Venn diagram. A Venn diagram was used to describe the number of high-confidence host proteins screened by a proximity interactome assay. (**C**) The purple part represents the overlapping host protein associated with NiV F and G proteins. (**D**) GO enrichment analysis. The histogram is used to depict the GO functional attributes of the 1996 high-confidence host proteins that interact with NiV F protein. (**E**) KEGG enrichment analysis. The histogram is used to depict the KEGG functional attributes of the top 25 KEGG pathways of 1996 high-confidence host proteins that interact with NiV F protein. (**F**) GO enrichment analysis. The histogram is used to depict the GO functional attributes of the 1524 high-confidence host proteins that interact with NiV G protein. (**G**) KEGG enrichment analysis. The histogram is used to depict the KEGG functional attributes of the top 25 KEGG pathways of 1524 high-confidence host proteins that interact with NiV G protein. (**H**) GO enrichment analysis. The histogram is used to depict the GO functional attributes of the 863 overlapping host proteins that interact with both NiV F and G proteins. (**I**) KEGG enrichment analysis. The histogram is used to depict the KEGG functional attributes of the top 25 KEGG pathway of 863 high-confidence host proteins that interact with both NiV F and G proteins.

**Figure 2 ijms-25-04112-f002:**
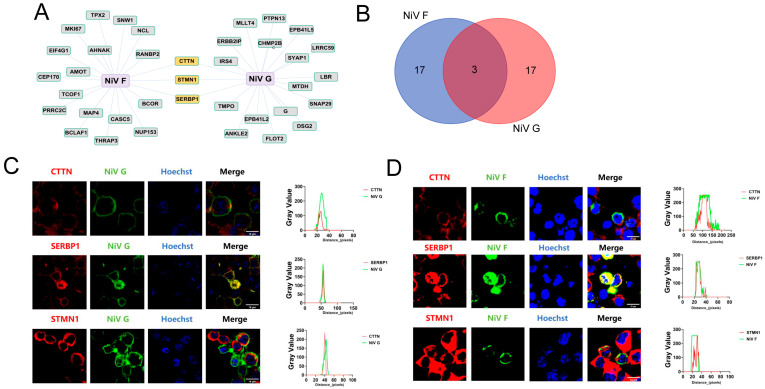
Analysis and identification of CTTN, SERBP1, and STMN1. (**A**) Visualization of top 20 interacting proteins. (**B**) Venn diagram. The Venn diagram depicts the number of overlapping host proteins with top 20 high-confidence proteins that interact with NiV F and G proteins. (**C**,**D**) Colocalization analysis. Myc-tag antibody, CTTN-, SERBP1-, and STMN1-specific antibodies were used as primary antibodies, FITC-anti-mouse and Cy3-anti-rabbit were used as secondary antibodies, Hoechst was applied for nuclear staining, and ImageJ software (ImageJ 1.53e, Java 1.8.0_172) was applied to analyze colocalization of CTTN, SERBP1, STMN1 with NiV F protein or NiV G protein according to the images taken by a confocal microscopy.

**Figure 3 ijms-25-04112-f003:**
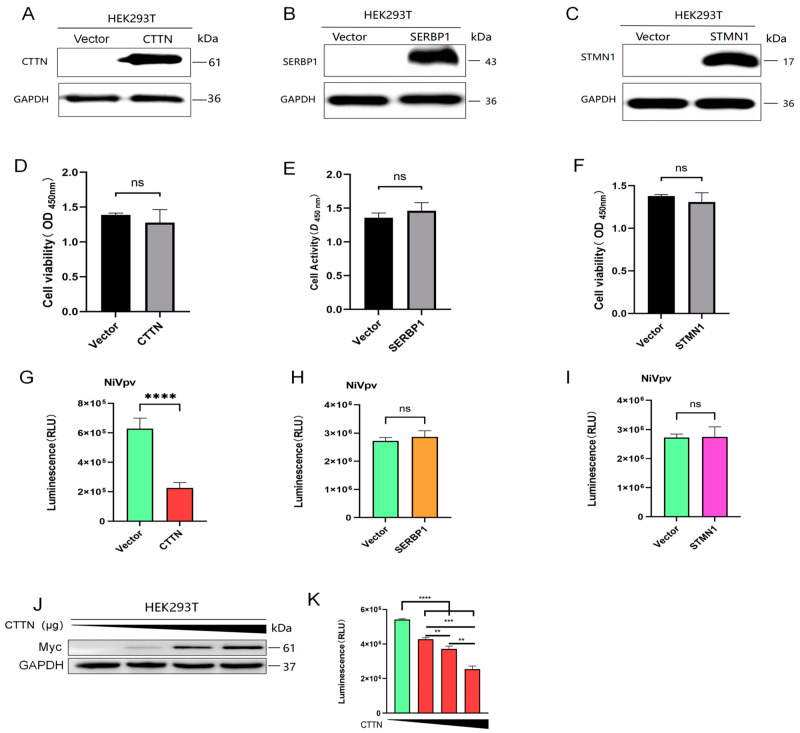
Overexpression of CTTN can inhibit NiVpv-infected cells in a dose-dependent manner. (**A**–**C**) Western blot assay. Western blots were conducted using anti-myc, anti-SERBP1, and anti-STMN1 antibodies as primary antibodies, and anti-mouse IgG or anti-rabbit IgG HRP conjugated antibodies as the secondary antibodies. (**D**–**F**) Cell viability analysis. Cell viability analysis was used to detect the effects of CTTN, SERBP1, STMN1 recombinant plasmids, and transfection reagents on cell viability. (**G**–**I**) Luciferase assay. Overexpression of CTTN inhibited the infection of NiVpv, while overexpression of SERBP1 and STMN1 had no significant effect on NiVpv infection. ns *p* > 0.05; **** *p* < 0.0001 (*t*-test), *n* = 4. (**J**) Western blot assay. Western blots were conducted using an anti-CTTN antibody as the primary antibody and an anti-rabbit IgG HRP conjugated antibody as the secondary antibody. (**K**) Luciferase assay. ** *p* ≤ 0.05; *** *p* ≤ 0.001; **** *p* ≤ 0.0001 (*t*-test).

**Figure 4 ijms-25-04112-f004:**
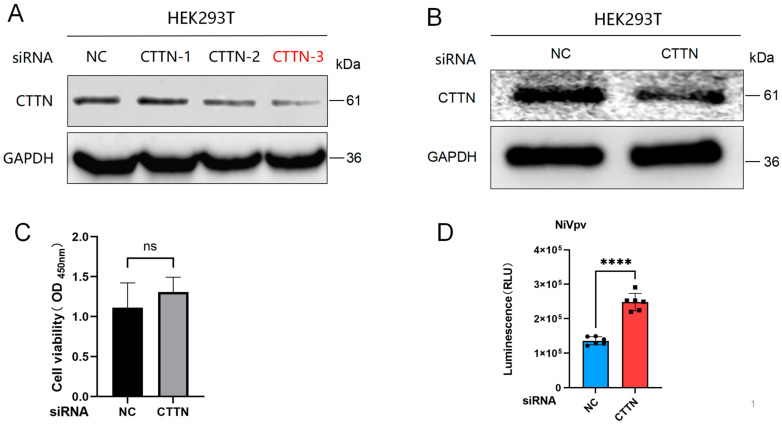
Knockdown of CTTN significantly inhibited NiVpv infection HEK293T cells. (**A**,**B**) Western blot assay. Western blots were conducted using an anti-CTTN antibody as the primary antibody and an anti-rabbit IgG HRP conjugated antibody as the secondary antibody. (**C**) Cell viability assay. The results showed that transfection of siRNA for 24 h had no significant effect on cell viability, ns *p* > 0.05 (*t*-test), *n* = 4. (**D**) Luciferase assay. Luciferase was used to detect the infection titers of NiVpv. Compared to the siNC group, knockdown CTTN could promote the infection of NiVpv significantly, **** *p* < 0.0001 (*t*-test), *n* = 6.

**Figure 5 ijms-25-04112-f005:**
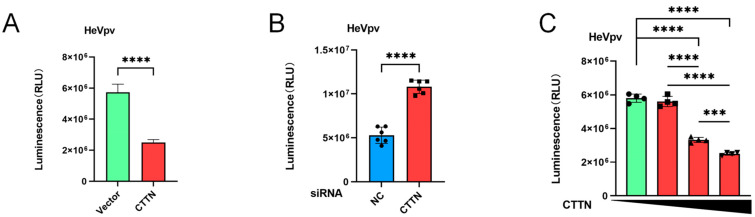
Knockdown and overexpression of CTTN can inhibit NiVpv-infected cells in a dose-dependent manner. (**A**) Luciferase assay was used to verify the effects of overexpression of CTTN on the infection of HeVpv infection, *p* < 0.0001 (*t*-test), *n* = 6. (**B**) Luciferase assay was used to verify the effects of knockdown of CTTN on the infection of HeVpv, *p* < 0.0001 (*t*-test), *n* = 6. (**C**) Luciferase assay was used to verify the effects of gradient overexpression of CTTN on the infection of HeVpv. *** *p* ≤ 0.001; **** *p* ≤ 0.0001, (*t*-test), *n* = 4.

## Data Availability

All data generated or analyzed during this study are included in this published article. The mass spectrometry proteomics data of proximity labeling have been deposited to iProX with accession number IPX0007741000 and can be archived through accession number PXD048109 in ProteomeXchange.

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
