# Peer review of "Interaction of Nipah Virus F and G with the Cellular Protein Cortactin Discovered by a Proximity Interactome Assay"

_ijms, 2024, doi:10.3390/ijms25074112_

Round 1

Reviewer 1 Report

Comments and Suggestions for Authors

The manuscript concerns an interesting health issue: Nipah virus biology. The Authors focused on the identification of cellular proteins that interact with the virus's F and G proteins.

However, the manuscript requires at least a few major additions and corrections.

Major comments:

  • Information must be provided on where raw data was deposited (which database). This information is necessary;

  • Methods are not sufficiently described, or even some methods are not described, e.g.:

    • Proximity labeling, the description is concise, and there are no references to the method.

    • More information is needed on the software and databases used for in silico data analysis obtained from the proximity labeling method.

    • There is no information/description on immunolocalization;

  • What post-tests were used for statistical analysis?

In general, the methods' descriptions do not reliably enable reproducing and evaluating the experiments.

  • RSV is included to Pneumoviridae;

Why was a pseudovirus used instead of an actual virus?

Why HEK293T cells were used?

Examples of minor comments concerning methods:

  • Which bacterial species are concerned with the description of Trans-T1?

  • What concentration of plasmids and siRNAs were used for transfection?

  • What microplate reader was used?

  • What concentrations of antibodies were used for Westen-blot?

  • What is the source of secondary antibodies for Western blot and immunolocalization?

Other minor comments:

  • Figure 1 is not readable, and the legend of Figure 1 is not correct. This is not an analysis of the database. I suppose that is the analysis of the results obtained by the Authors.

  • Editiorial errors;

  • Authors should correct English, e.g., lines 120 – 1128 contain one sentence.

Comments on the Quality of English Language

Extensive editing of English language required. For example, lines 120-128 contain one sentence.

Reviewer 2 Report

Comments and Suggestions for Authors

The study investigates the possible interactions of Nipah virus F and G with the cellular protein Cortactin.

I have marked some issues that need addressing, before the manuscript can be considered again.

1.      The Introduction does not provide adequate information to initiate readers into the study. The authors should expand and also should include details more specific to their study rather than just generalities.

2.      The objectives of the study have not defined clearly.

3.      HEK293T cells: please provide the origin of these cells.

4.      Please describe clearly how repetitions were performed in each experiment and in each assay.

5.      Analysis: why using parametric tests? No evidence regarding normality of the data has been presented. Please correct.

6.      The Discussion is shallow and does not cover fully the findings of the study. Please expand and please go into greater depth.

7.      Some recent references on the topic should be discussed regarding their similarity to the results of the authors.

8.      Please explain in the Discussion a) the target readership for this manuscript and b) whether this idea can be commercialized.

Overall: as it is, this is a weak manuscript; extensive improvement is needed, which should be followed by further peer-reviewing.

Round 2

Reviewer 2 Report

Comments and Suggestions for Authors

The authors have made a good job to improve their manuscript and have addressed all the previous comments.

Before final acceptance, please have a thorough reading to correct some misspellings and syntax errors throughout the manuscript. Also, some references are badly located at the wrong sentence and need to be moved at the end of the relevant paragraphs.
Then, the manuscript will be acceptable.
